# Snow Cover Phenology in Xinjiang Based on a Novel Method and MOD10A1 Data

Qingxue Wang [1,2], Yonggang Ma [2,3,4,5,*] and Junli Li [6,7,8]

1   College of Geography and Remote Sensing Sciences, Xinjiang University, Urumqi 830046, China
2   Xinjiang Key Laboratory of Oasis Ecology, Xinjiang University, Urumqi 830046, China
3   College of Ecology and Environment, Xinjiang University, Urumqi 830046, China
4   Xinjiang Jinghe Observation and Research Station of Temperate Desert Ecosystem, Ministry of Education, Urumqi 830046, China
5   Key Laboratory of Oasis Ecology of Education Ministry, Urumqi 830046, China
6   State Key Laboratory of Desert and Oasis Ecology, Xinjiang Institute of Ecology and Geography, Chinese Academy of Sciences, Urumqi 830011, China
7   University of Chinese Academy of Sciences, Beijing 100049, China
8   Key Laboratory of GIS & RS Application, Xinjiang Uygur Autonomous Region, Urumqi 830011, China
\*   Correspondence: mayg@xju.edu.cn

**Abstract:** Using Earth observation to accurately extract snow phenology changes is of great significance for deepening the understanding of the ecological environment and hydrological process, agricultural and animal husbandry production, and high-quality development of the social economy in Xinjiang. Considering snow cover phenology based on MODIS product MOD10A1 data, this paper constructed a method for automatically extracting key phenological parameters in Xinjiang and calculated three key phenological parameters in Xinjiang from 2001 to 2020, including SCD (snow cover duration), SOD (snow onset date), and SED (snow end date). The daily data of four field camera observation points during an overlapping period from 2017 to 2019 were used to evaluate the snow cover phenological parameters extracted by MOD10A1, and the mean absolute error (MAE) and root mean square error (RMSE) values were 0.65 and 1.07, respectively. The results showed the following: 1. The spatiotemporal variation in snow phenology was highly altitude dependent. The mean gradients of SCD in the Altai Mountains, Tienshan Mountains, and Kunlun Mountains is 2.6, 2.1, and 1.2 d/100 m, respectively. The variation trend of snow phenology with latitude and longitude was mainly related to the topography of Xinjiang. Snow phenological parameters of different land-use types were different. The SCD values in wasteland were the lowest and the SED was the earliest, while forest land was the first to enter SOD accumulation. According to the study, the mean annual values of SCD, SOD, and SED were 25, 342 (8 December), and 51 (8 February) as day of year (DOY), respectively. The snow cover area was mainly distributed in the Altai Mountains, Junggar Basin, Tienshan Mountains, and Kunlun Mountains. 2. The variation trend and significance of snow cover phenological parameters in different regions are different, and the variation trend of snow cover phenological parameters in most regions of Xinjiang is non-significant.

**Keywords:** snow phenology; Xinjiang; MODIS; snow cover; MK-Test

## 1. Introduction

As the most widely distributed factor of the cryosphere and the most sensitive to interannual variation, snow cover is an important factor affecting the global climate [1]. Like in most arid and semi-arid zones, snowmelt provides an important proportion of the fresh water in Xinjiang, China [2]. Snow cover phenology, including SCD (snow cover duration), SOD (snow onset date), and SED (snow end date), is an important index to measure the characteristics of snow cover change, which has an important impact on the seasonal changes of terrestrial ecosystems, especially on the phenology of alpine

vegetation [3,4]. Early snow melt can lead to significant changes in the timing and flow of spring snowmelt runoff, potentially increasing the incidence of events such as floods and droughts [5]. During the winter, snow helps agriculture by trapping heat from the surface and thereby protecting crops (mainly winter wheat) from low-temperature frosts [6]. Therefore, taking Xinjiang as the research area, studying the spatiotemporal variation in snow phenology is of great significance for regional and even global climate change, water resource management, vegetation growth [7], sustainable development, and the prediction of catastrophic climate events [8].

Since the last century, a significant amount of research has been carried out on snow information extraction using remote sensing technology on the regional and even global scale [9–13]. Remote sensing data, with the characteristics of multi-scale, multi-temporal, multi-spectral, and multi-level, provide a high-quality data source for the study of snow cover in high-altitude mountainous areas [14–18]. The result shows that SCD has decreased in the global mountains. It is necessary to understand the snow cover dynamics and prepare for the future changes of global snow cover resources [19,20]. Most research on the progress of snow cover in the Northern Hemisphere [21–28] has found that the SCD has shortened in most parts of the Northern Hemisphere. In the Arctic, the snow cover in June is declining at a rate of 13.6% per decade. In North and South America, the SCD peak time has moved 0.6 days earlier every 10 years [29–32]. In Eurasia, the spatial range of snow cover in spring and summer has decreased significantly, and the time when snow begins to disappear in spring has significantly advanced [33–35].

In recent years, the information sources of snow cover have diversified, and station and satellite data have become important resources. Remote sensing data and ground observation data are used to analyze the distribution characteristics and spatiotemporal characteristics of snow cover [36–38]. The snow product synthesized by MODIS provides long time series, high precision spatial snow cover distribution information, and spatial surface temperature information and has the advantages of convenient and quick data acquisition, making the data one of the best sources for large-scale snow cover research. These data have been widely used in snow cover research [39]. The accuracy of the surface albedo products of MODIS V005 and V006 are compared globally using the ground station data, and it is concluded that the overall accuracy of V006 is higher than that of V005 [40,41]. An update of the MOD10A1 snow product may supply a more accurate option for this field. Liu [42] took three research areas in southwest Xinjiang as examples to evaluate the accuracy of MOD10A1 and found that the average overall accuracy of MOD10A1 reached 82%. Chen and his team [43] obtained that the average time of SCD, SOD, and SED of snow phenological parameters in the Tianshan Mountains was 113.9, 331.2, and 81, and the average gradient of SCD, SOD, and SED with elevation was 6.0, −2.55, and 3.44 d/100 m, respectively. With high accuracy, MOD10A1 can extract snow information in different terrains. The MODIS product has provided clues that assist in understanding the dynamics of snow change. Compared with other long time series data, MODIS has a higher resolution, and the advantage of long time series is mostly used by researchers.

The above studies mainly focus on the changes of snow cover extent and snow cover days [44–49], while less attention is paid to the spatiotemporal changes of the onset and melting time of snow cover. In addition, different geographical locations have different snow cover conditions, but most studies only focus on snow cover information in mountains [50–52] or large spatial patterns [53]. Therefore, it is necessary to analyze the snow cover phenological pattern and trend change in different geographical units in the whole Xinjiang region. However, there are difficulties in the accurate calculation of SOD and SED in snow phenological parameters, especially the influence of instantaneous snow cover in early winter. Moreover, the definition of snow phenological parameters is different, and the extraction methods are different. The objective of this study is to use a new method to calculate the snow phenological parameters of SCD (snow cover duration), SOD (snow onset date), and SED (snow end date) from 2001 to 2020. We use the daily data of four field camera observation points to verify and evaluate the snow phenological parameters

extracted by MOD10A1. Then, the phenological pattern and trend changes in different elevation regions and geographical units in the entire Xinjiang region were analyzed.

## 2. Materials and Methods

### 2.1. Study Area

Xinjiang, the largest province in China by land area, is located between 73°40′~96°18′E and 34°25′~48°10′N. It is in the hinterland of Eurasia. With complex terrain and climate patterns, Xinjiang boasts a variety of unique landforms and ecosystems of diverse types [54]. The altitude is −190~7538 m, with major differences in altitude. The high altitude and unique geographical location provide good climatic conditions for the development of snow and glaciers in the mountainous areas of Xinjiang. A change of snow cover is related to the life, industry, and irrigation water in Xinjiang [55]. According to the geographical topography and ecological environment of the study area, regional differences of snow phenological parameters were studied (Figure 1).

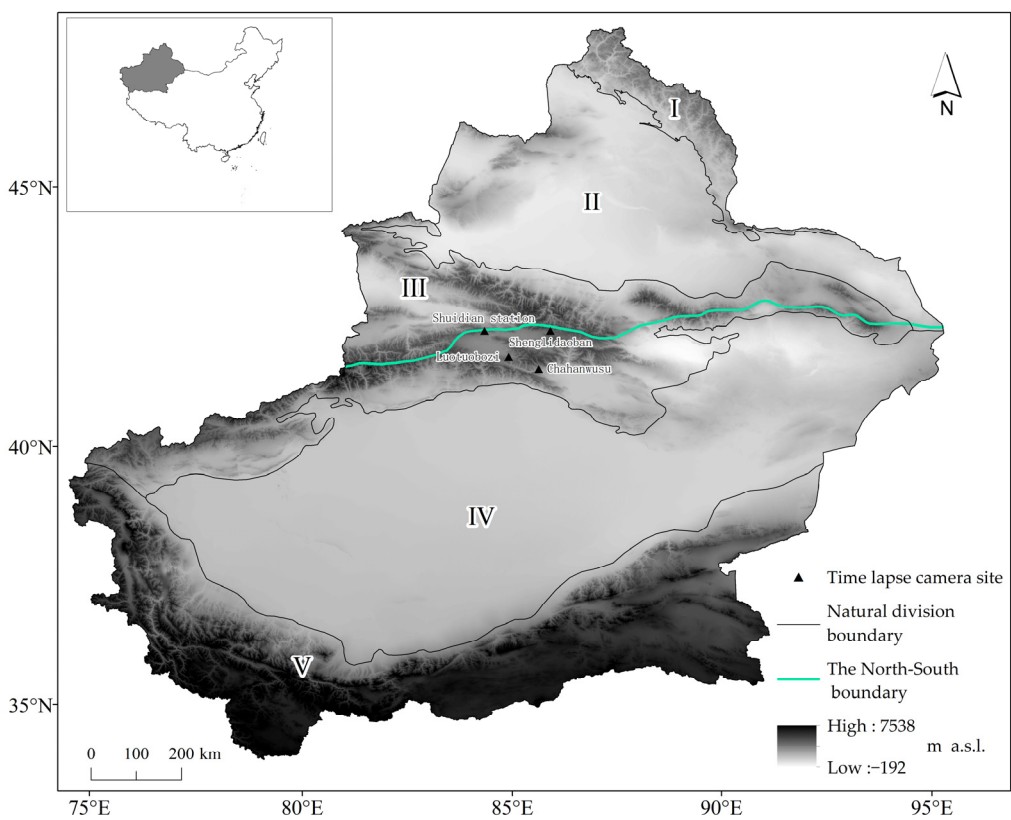

**Figure 1.** Study area and time−lapse camera stations: (I) Altai Mountains; (II) Junggar Basin; (III) Tienshan Mountains; (IV) Tarim Basin; (V). Kunlun Mountains.

### 2.2. Data Sets

Snow cover data set: The snow cover data set considers the spatial and temporal resolution of various remote sensing data currently available. The MOD10A1 V6 Snow Cover Daily Global 500 m from 2001 to 2020 was used [56] (data from https://www.earthdata.nasa.gov/, accessed on 4 July 2020). The Xinjiang region involved MOD10A1 images (from h23v04, h23v05, h24v04, h24v05, h25v04, h25v05) and was used as the basic data for snow cover phenological information extraction. QGIS, ArcGIS, and Python were used to pretreat daily snow cover products in the study area for format conversion, projection conversion, and clipping snow cover phenology data extraction.

Time-lapse camera site data: Our research group established four field camera observation points in the middle part of the Xinjiang mountains during 2017 to 2019, which provided important data support for the accuracy verification of snow products in moun-

tainous areas. The accuracy of MOD10A1 products in the middle Tianshan Mountains of Xinjiang was evaluated by the field camera image data.

DEM: To explore the relationship between snow phenology and elevation variations, the Digital Elevation Model (DEM) data were derived from the SRTM (Shuttle Radar Topography Mission). Spatial resolution was 90 m and data format was Geo-TIFF [57]. The absolute vertical height accuracy was reported to be less than 16 m, while the horizontal accuracy was 20 m [58]. The advantage of SRTM is its consistency in data collection and analysis; thus, data quality is homogeneous. The resolution of DEM (90 m) data was aggregated to 500 m to match the spatial resolution of the snow cover data.

MCD12Q1: The MODIS Land Cover Type Product (MCD12Q1) provides a suite of science data sets (SDSs) that map global land cover at 500 m spatial resolution at annual time steps for six different land cover legends; the MCD12Q1 International Geosphere-Biosphere Programme (IGBP) includes a legend and class descriptions. The influence of underlying cover on snow phenological parameters was discussed, and 17 main land cover types were combined into several categories for unified calculation (15 snow and ice deletions) (Table 1). Using the MCD12Q1_v06 IGBP data set from 2001 to 2020, ARCGIS was used to analyze the land-use types that did not change during 2001 to 2020, and these unchanged land data and snow phenological parameter data were used for statistical analysis.

**Table 1.** Classification of the main land cover types.

| MCD12Q1_v06 IGBP | Data Sets | Classification |
|---|---|---|
| 1- Evergreen needleleaf forest<br>2- Evergreen broadleaf forest<br>3- Deciduous needleleaf forest<br>4- Deciduous broadleaf forest | 5- Mixed forests<br>6- Closed shrubland<br>7- Open shrublands<br>8- Woody savannas<br>9- Savannas | Forest |
| 10- Grasslands | | Grasslands |
| 12- Croplands<br>14- Cropland/natural vegetation mosaic | | Croplands |
| 11- Permanent wetlands | | Permanent wetlands |
| 13- Urban and built-up | | Urban and built-up |
| 16- Barren or sparsely vegetated | | Barren or sparsely vegetated |
| 17- Water | | Water |

*2.3. Snow Cover Phenological Parameter Extraction Method*

2.3.1. Three Published Methods for Snow Cover Phenology Extraction

The definitions of snow phenological parameters are different, and the extraction methods are also different. Table 2 summarizes three common methods. The first two methods are to obtain snow parameters from remote sensing images, and the last one is the method traditionally used in China.

In these three methods of snow cover phenology extraction, Wang and Xie's method set two special time periods to calculate the SOD and SED. If there is snow cover outside the specified time, the snow cover date in areas cannot be detected. Their results showed that the overall agreement between the SCD and site observations was 90%, and the SOD or the SED had a good agreement, with a mean of one week earlier or one week later, respectively. However, after GAO's evaluation, there are some differences in snow phenological parameters in complex study areas, mainly because the SOD and the SED in snow could not be found outside the set site date. These assumptions are applicable to study areas in special areas but need to be carefully considered for study areas with transitional complex terrain and high altitude.

**Table 2.** Snow cover phenological parameter extraction methods.

| Method | SCD | SOD | SED | Description |
|---|---|---|---|---|
| Wang and Xie [59] | $SCD = \sum_{0}^{N} H(D_i - 50)$<br>$N$ represents the number of days contained in a hydrological year; $D_i$ is the snow cover fraction (%) in a pixel $(0 \leq D_i \leq 100)$, and the H function equals 0 (1) for negative (positive) arguments. | $SOD = D_1 - SCD'_1$<br>$SCD'_1$ represents snow-covered days within the period from 1 September to 20 January of the following year. | $SED = D_2 + SCD'_2$<br>$SCD'_2$ indicates snow-covered days within the period from 21 January to 31 August of the same year. | The algorithm described by Wang and Xie was used to determine the snow cover phenological parameters, which could be slightly modified. This method explains the early and late-season brief snow events. |
| GAO Y [60] | Method 1 for Cloud-Free Snow Cover: SCD is simply calculated as the total snow cover days for each pixel in one hydrological year. Method 2 for Low Cloud Snow Cover: SCD is the number of snow cover days, includes all days within the maximum continuous snow-covered period and snow cover days in transient snowfall events. | Method 1 for Cloud-Free Snow Cover: the first date when the pixel value is snow for 14 consecutive days. Method 2 for Low Cloud Snow Cover: the time series is searched to satisfy two conditions. | Method 1 for Cloud-Free Snow Cover: the first date when the pixel value is land for 14 consecutive days. Method 2 for Low Cloud Snow Cover: a pixel with SCOD will be searched from its SCOD to the last day of one hydrological year to satisfy two conditions. | GAO Y has two methods for calculating snow cover parameters from no-cloud (low cloud) snow cover maps. In the case of low cloud, the new method for obtaining snow cover parameters by setting different conditions at different times is generally based on 14 consecutive days of snow cover or 14 consecutive days as ground. |
| Phenological Observation Methods in China [61] | - | First snow cover: The date when snow cover is first seen on the ground (half of the ground near the phenological observation point is snow cover). | Snow cover melting: On flat ground, the date when the snow cover first melts to reveal the ground and the date when it completely melts (in low recesses) to reveal all the ground. | Select the day of complete snow melt in the Natural Phenology Annual Snow Record. |

GAO's Method 2 was applied to the low cloud MODISMC8 map to obtain snow cover parameters by setting different conditions in different periods. In the Pacific Northwest of the United States, the differences of SCD, SOD, and SED were 70%, 72%, and 55%, respectively. The performance of this method is better than GAO's Method 1 and Wang and Xie. However, the accuracy of the Wang and Xie methods was higher in northern Xinjiang.

The method of Phenological Observation Methods in China is the traditional method snow parameters are obtained by field observation. However, the measurement sites cannot provide large-scale regional and global scale data because their spatial density is low, the sites are not continuous, even in hard-to-reach areas, and the sites are mostly located in human activity areas and low-altitude zones, so there are many gaps in the data of periods and regions where no phenological observation sites have been established. Therefore, different regions adapt to different snow phenology extraction methods, so we put forward a new method to calculate snow phenology by taking advantage of the complex geographical terrain of Xinjiang and then verified our method.

### 2.3.2. A New Method for Extracting Snow Phenological Parameters

Based on the temperate continental climate and unique topographic conditions in Xinjiang, there are temperature differences in the four seasons, with obvious seasonality. Combining the changes of different geographical regions, this study proposed an algorithm suitable for snow phenology retrieval in Xinjiang. Snow cover phenological parameters were extracted in a calendar year (starting in January and ending in December), which was convenient for quantification and analysis of the annual snow accumulation period. A hydrological year is defined as 1 August of the current year to 31 July of the following year. After the snow cover phenological parameters were obtained, the changes of snow cover phenological parameters were converted into hydrological year dates for analysis.

SCD: In the NDSI_Snow_Cover data set, records with pixels ranging from 1 to 100 are recorded as snow, and other values are recorded as free snow. In the annual pixel time series, SCD is the number of pixels whose pixel value is 1–100.

$$SCD = \sum_{0}^{N} D_i \tag{1}$$

SOD: In the pixel-by-pixel time series, the time series of each pixel was extracted from the calendar year from 1 January to 31 December. In the obtained time series of snow cover, the larger of the values of the two snow cover days (Julian days) with the biggest difference between the two days before and after the snow cover days is the SOD.

SED: In the pixel-by-pixel time series, the time series of each pixel was extracted from the calendar year from 1 January to 31 December. In the obtained time series of snow cover, the smaller of the values of the two snow cover days (Julian days) with the biggest difference between the two days before and after the snow cover days is the SED (Figure 2).

$$gap(x) = index_x - index_{x-1} \ x \in (2, SCD) \tag{2}$$

$$x' = arg \ max \ gap(x) \tag{3}$$

$$SOD = index_{x'} \tag{4}$$

$$SED = index_{x'-1} \tag{5}$$

**Figure 2.** Examples of the SOD and SED defined (the number represents the Julian day with snow cover in the calendar year).

In the formula, $index_x$ is the pixel time series with snow cover. Figure 2 shows that in the time series $index_x$ of snow cover, day 96 (6 March) is the SED, and day 308 (4 November) is the SOD.

Snow phenological parameters of three different extraction methods were compared in detail (Figure 3) for January 2016 to 31 December 2017. The new method includes transient snow cover in autumn and winter as well as short periods of snow accumulation.

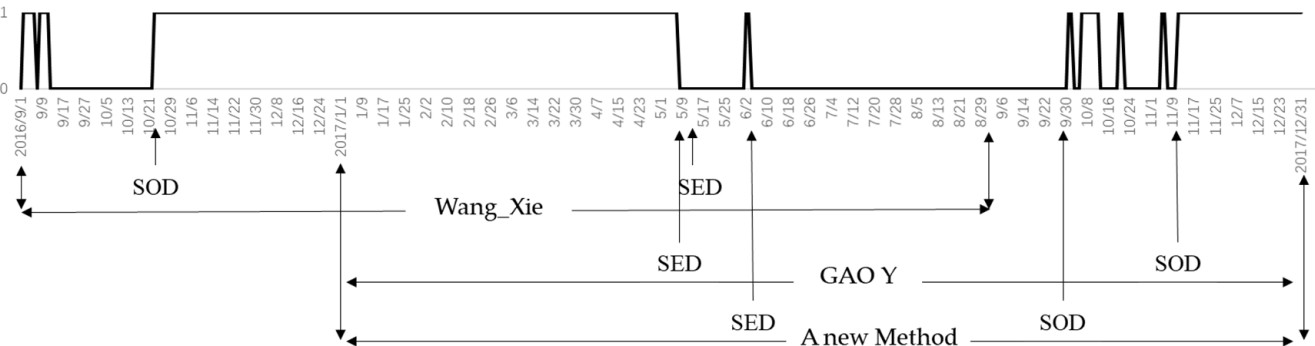

**Figure 3.** Comparison of three methods to extract the snow phenology parameters (0 means free snow, 1 means snow).

*2.4. Accuracy*

The overall accuracy of snow cover phenological parameters extracted by MOD10A1 was evaluated by using time-lapse camera photo data in the field. The mean absolute error (*MAE*) and root mean square error (*RMSE*) were calculated by the following formula:

$$MAE = \frac{\sum_{i=1}^{n}|c_i - s_i|}{n} \tag{6}$$

$$RMSE = \sqrt{\frac{\sum_{i=1}^{n}(c_i - s_i)^2}{n}} \tag{7}$$

where *n* represents the number of pixels in the comparison, $C_i$ is snow cover phenological ("calculated value") extracted for the MOD10A1 product, and $S_i$ is the truth value of time-lapse camera data.

From 2017 to 2019, our research group set up four field camera observation sites in the middle part of the Tianshan Mountain area in Xinjiang (Figure 1). These observation sites were characterized by high altitude and less interference from human activities, which provides important data support for accuracy verification of snow products in mountainous areas. First, the camera photos were manually interpreted to list the days when snow remained after 10 a.m. each day. Then, the snow phenological data extracted by the method of extracting snow phenological parameters in this paper were used as the true value, and the snow phenological parameters extracted by MOD10A1 were evaluated. According to Table 3, snow phenological parameters extracted by MOD10A1 in this paper differ from field camera data by 0 to 3 days, which is caused by cloud influence, which is more accurate than Wang and Xie's method. The calculated mean absolute error (*MAE*) and root mean square error (RMSE) values were 0.65 and 1.07, respectively.

**Table 3.** Comparison between camera data and MOD10A1 snow cover phenological parameters. (Unit: DOY).

| Year | Station Name<br>Unit: m a.s.l.<br>Type | Luotuobozi<br>(2395.19)<br>SOD | SED | Shuidianzhan<br>(2955.58)<br>SOD | SED | Shenglidaoban<br>(3317.75)<br>SOD | SED | Chahanwusu<br>(1962.19)<br>SOD | SED |
|------|------|------|------|------|------|------|------|------|------|
| 2017 | MOD10A1 | 275 | 159 | 268 | 159 | 275 | 160 | 0 | 0 |
|  | time-lapse camera | 274 | 160 | 267 | 159 | 275 | 160 | 0 | 0 |
| 2018 | MOD10A1 | 268 | 146 | 255 | 145 | 244 | 148 | 292 | 39 |
|  | time-lapse camera | 268 | 146 | 255 | 146 | - | - | 291 | 39 |
| 2019 | MOD10A1 | 313 | 128 | 253 | 197 | 255 | 139 | 33 | 30 |
|  | time-lapse camera | 311 | 128 | 253 | 198 | - | - | 33 | 30 |

*2.5. Trend Analysis Method*

Sen's [62] slope estimation method and the Mann–Kendall [63] test method were used to explore the variation trend and significance of the snow phenological parameter amplitude.

Sen's slope estimation is a robust non-parametric statistical method for trend calculation. This method is computationally efficient and insensitive to measurement errors and outliers, so it is used in trend analysis of long time series data.

$$\beta = Median \frac{x_j - x_i}{j - i}, \forall j > i \tag{8}$$

where $x_j$ and $x_i$ are time series data. When $\beta$ is greater than 0, the time series shows an upward trend. A $\beta$ less than 0 indicates a decreasing trend in the time series.

The Mann–Kendall test is a non-parametric trend analysis method. Compared with other parameter testing methods, the samples do not need to follow a certain distribution and are less disturbed by outliers, which is more suitable for ordinal variables.

The Mann–Kendall test has been widely used in the study of hydrological and meteorological trend change and is widely used in the analysis of time series data in the hydrological, meteorological, and ecological fields [64].

Time trends of statistical significance by z values assume that when the acuity is 1.28, 1.64, and 2.32, respectively, the confidence level is 90%, 95%, and 99% of the test of significance. For the time series variables $x_1, x_2, \ldots, x_n$, $n$, is the length of the time series, and the test statistic $S$ is defined as:

$$S = \sum_{i=1}^{n-1} \sum_{j=i+1}^{n} sgn(x_j - x_i) \tag{9}$$

$$sgn(x_j - x_i) = \begin{cases} 1 & x_j > x_i \\ 0 & x_j = x_i \\ -1 & x_j < x_i \end{cases} \tag{10}$$

If a data set displays a consistently increasing or decreasing trend, $S$ will be positive or negative, respectively, with a larger magnitude indicating the trend is more consistent in its direction.

The variance of the Mann–Kendall statistic is:

$$Var(S) = \frac{n (n-1)(2n+5)}{18} \tag{11}$$

where $n$ is the number of data points.

The test statistic $Z$ is calculated as follows:

$$Z = \begin{cases} \frac{S-1}{\sqrt{Var(S)}} & S > 0 \\ 0 & S = 0 \\ \frac{S+1}{\sqrt{Var(S)}} & S < 0 \end{cases} \tag{12}$$

## 3. Results

To verify the applicability of the new method, we used it to obtain snow phenological parameters in Xinjiang during 2001~2020 and studied the spatial distribution and change trend of snow phenology in Xinjiang.

*3.1. Spatial Distribution Pattern of Snow Cover Phenology in Xinjiang*

To obtain the detailed spatial and temporal distribution pattern, MOD10A1 snow cover data were used to obtain the long-term series variation in snow cover in Xinjiang from 2001 to 2020 (Figure 4). Considering the different climatic conditions in northern and southern Xinjiang, the snow cover change in northern and southern Xinjiang was compared

with that in the whole of Xinjiang. The results showed that the snow cover area in the study area had significant seasonal changes, with an obvious snow accumulation period and melting period. The proportion of snow cover reached the maximum in winter, decreased with the coming of spring, reached the minimum in summer, gradually increased after the beginning of autumn, and finally reached the maximum in winter. The annual variation in snow cover in northern Xinjiang was significantly higher than that in southern Xinjiang, and the overall fluctuation trend remained the same. From 2001 to 2020, the maximum annual snow cover in the whole of Xinjiang was 20–30%, and the occurrence time was mostly from November to March of the next year. The minimum value range was 1.7–2.6%, and the occurrence time was mostly from July to August. In addition, the fluctuation of the maximum snow cover was more violent than that of the minimum.

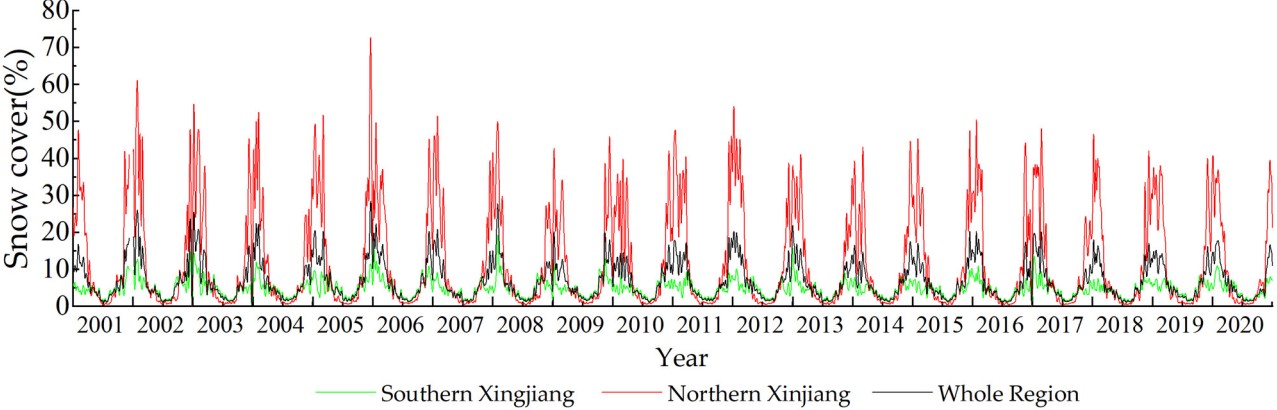

**Figure 4.** Interannual change of the snow cover fraction of Xinjiang (includes northern region and southern region) from 2001 to 2020.

Snow phenological parameters in the study area showed strong spatial differences (Figure 5). The mean annual values of SCD, SOD, and SED were 25, 342 (8 December), and 51 (8 February) in the day of the year (DOY), respectively. Snow cover areas were distributed in the Altai Mountains, Tarim Basin, Tianshan Mountains, and Kunlun Mountains, accounting for 80.72% of the total area of the study area (Figure 5a). The area with an SCD of more than 100 days accounted for 5.15% of the total area of the study. In addition, 70.98% of the areas with an SCD of less than 30 days were distributed in the Junggar Basin and the low-altitude areas of the Tarim Basin. Half of the areas in the Tarim Basin had no snow cover and half had an SCD of less than 30 days. According to the diagram of the beginning date of snow cover (Figure 5b), the terrain is like that of Xinjiang overall, and the boundary between the three mountain areas and the basin is obvious. The beginning date of snow cover in the mountain area is early September, which is earlier than that in the Junggar Basin, and the SOD in the Junggar Basin mostly appears in November and later. According to the spatial distribution of the SED (Figure 5c), it was mainly concentrated from January to June and appeared in March or before in the two basins. High-altitude mountain areas are delayed until May and beyond.

The snow cover phenological parameters in the mountain areas showed an obvious elevation gradient (Figure 6). The mean gradient of SCD, SOD, and SED with elevation are 2.6 d, −1.7, and 2.2 d/100 m, respectively in the Altai Mountains, and the mean gradient of the SCD, SOD, and SED was 2.1 d, −1.7, and 3.2 d/100 m in the Tianshan Mountains, respectively. The SCD gradient in the Kunlun Mountains is relatively slow at 1.2 d/100 m, while the SOD and the SED gradients were −1.2 and 2.3 d/100 m, respectively (Figure 6c). With the increase of altitude, SCD increased of the region and advanced SOD and delayed SED were observed. However, with the change of altitude, the inflection point of SOD and SED in the Altai Mountains is 4000 m, the inflection point of SOD and SED in Kunlun Mountains is 4500 m, and the inflection point of SOD and SED in the Tianshan Mountains is 3500 m. This may be due to the influence of perennial snow and glaciers, which causes

the SOD to be delayed and SED to be advanced. This difference may be due to the decrease in water pressure and the strengthening of solar radiation in high-altitude areas, the faster increase in temperature than in low altitude areas, and the effect of wind transport.

**Figure 5.** Average snow cover phenology in Xinjiang during 2001–2020: SCD (**a**); SOD (**b**); and SED (**c**).

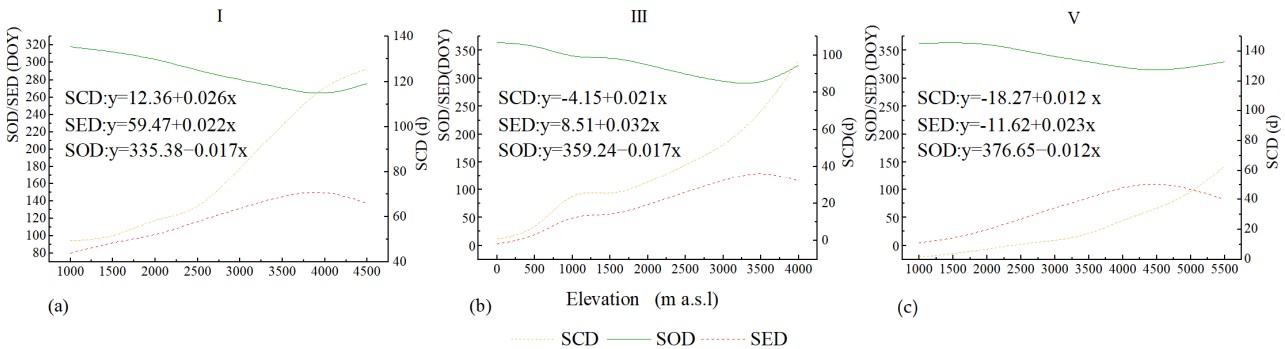

**Figure 6.** Variations of snow cover phenology with elevation in different subdivisions: (**a**) Altai Mountains; (**b**) Tienshan Mountains; (**c**) Kunlun Mountains.

The variation trend of snow phenological parameters with longitude in the study area is shown in Figure 7. Xinjiang has a vast area and a large latitude and longitude span. The snow phenological parameters fluctuate greatly with latitude and longitude changes. Two inflection points with large fluctuation were found in the analysis of longitude variation in the phenological parameters, which may be related to the geographical mountain conditions in the study area. In the analysis of latitude variation in phenological parameters, it was found that the SCD was depressed (low value) between 38° and 40°N, which may be because this area is the Tarim Basin, and the phenological parameters of snow cover were lower in the same latitude region, thus forming a depression area. Then, an inflection point appeared at 43°N, which was mainly due to the influence of the Tianshan Mountains. Figure 7a,b reflect the interannual variation in the SCD with higher latitudes in mountain areas, with the Altai Mountains > Tianshan Mountains > Kunlun Mountains. Figure 7c,d show that the SCD in the study area decreased with the increase in longitude, while the SOD was delayed, and the SED advanced.

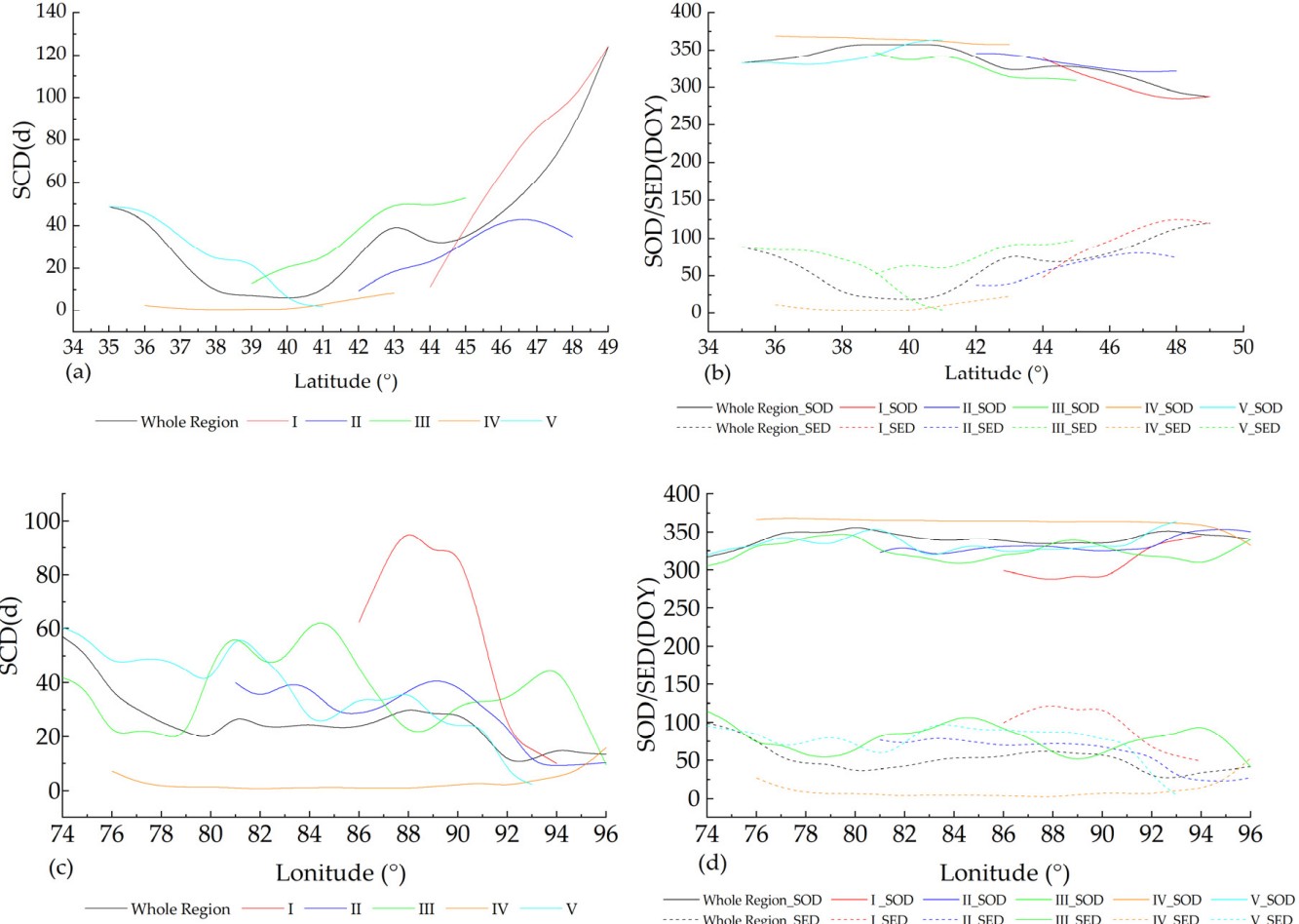

**Figure 7.** Variations of snow cover phenology with latitude and longitude: SCD (**a**) and SOD/SED (**b**) in latitude; SCD (**c**) and SOD/SED (**d**) in longitude.

Different land-use types have different solar radiation, which affects the change process of accumulation and ablation in this region. According to Figure 8a, the fluctuation in the SCD of different land-use types remained stable during the study period. The average SCD of barren land was at least 18 days, the average annual SCD of forest was 78 days, and the average annual SCD of grasslands was 49 days. Figure 8b analyzes the change characteristics of snow start date under different vegetation types (). The average SOD of forest land is the 311th day, which first enters the snow accumulation state, and the average SOD of wasteland is the 349th day, which finally enters the snow accumulation state. After

the SED (Figure 8c), the average SED of forest is the 93rd day of snow melting, which is later than the 87th day of the SED of grassland, and the earliest end time of snow cover in wasteland is the 40th day.

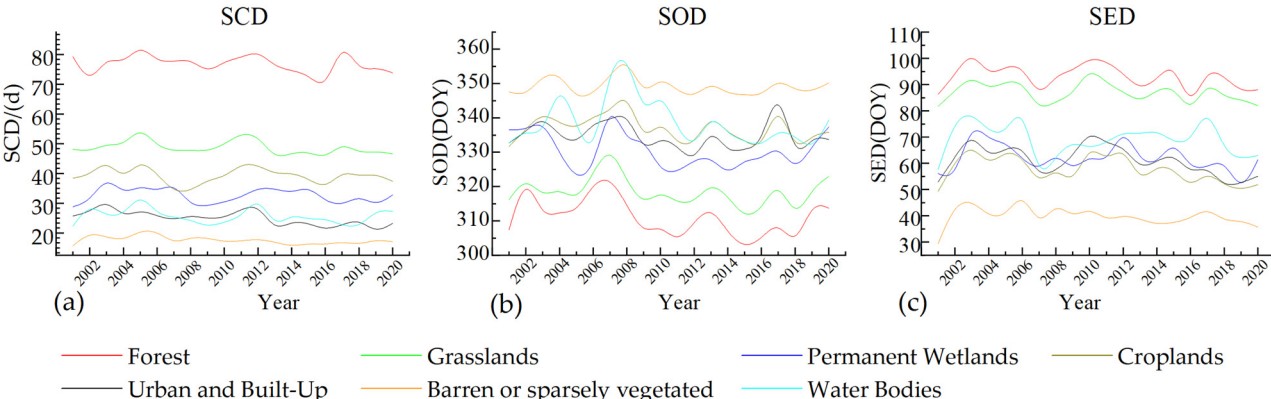

**Figure 8.** Variations of snow cover phenology under different land-use types: SCD (**a**); SOD (**b**); and SED (**c**).

### 3.2. Spatiotemporal Variation in Snow Phenology in Xinjiang

The change trend analysis and significance test of the SCD, SOD, and SED in the study area from 2001 to 2020 were carried out (Figure 9), and the trend change of snow phenology parameters in the Xinjiang region (Table 4) and the proportion of the area with a significant trend were calculated. The snow phenology in most areas showed a non-significant change (Figure 9). Some areas showed significant changes.

The SCD in most areas of the study area showed a non-significant shortening trend (Figure 9a,b), while 8.49% of Xinjiang showed a significant decrease in the Junggar Basin and southern Xinjiang, with a shortening rate of about 0.67 $d \cdot a^{-1}$ ($p < 0.05$). A total of 1.44% of Xinjiang had a significant lengthening trend, and the lengthening rate was about 0.97 $d \cdot a^{-1}$ ($p < 0.05$), distributed in the central Tianshan and Kunlun Mountains. In southern Xinjiang, 1.96% of the SCD showed a significant increase trend ($p < 0.05$). For the whole study area, there was an insignificant decrease in SCD at 0.12 $d \cdot a^{-1}$ (Table 4), which was related to the advance of the SED.

The SOD was significantly advanced in 1.75% of the Xinjiang region ($p < 0.05$) (Figure 9c,d) (Table 4), with an advance rate of 2.35 $d \cdot a^{-1}$, mainly distributed in the Altai Mountains, the Junggar Basin, the central Tianshan Mountains, and the area where the Kunlun Mountains meet the Tarim Basin (Figure 9c,d). The SOD was significantly delayed in 1.32% of the region ($p < 0.05$), with a rate of about 3 $d \cdot a^{-1}$, mainly distributed in the Kunlun Mountains, the northeastern Tarim Basin, and the eastern margin of the Junggar Basin. The SOD in the Tarim Basin and the Kunlun Mountains was postponed at the rates of 0.02 $d/a$ and 0.12 $d/a$, respectively, but the trend change was not significant. The rate of SOD in the Altai Mountains was 0.33 $d/a$ ahead of that in other regions

According to the trend and spatial distribution of SED significance (Figure 9e,f), the SED showed a significant advance in 2.38% of the Xinjiang region ($p < 0.05$) (Table 4), mainly distributed in the Altai Mountains area, the Junggar Basin area, the western part of the Tianshan area and Tarim Basin, the Kunlun Mountains area, and the Tarim Basin, with an advance rate of 2.47 $d \cdot a^{-1}$. The SED was significantly delayed in 1.01% of the Xinjiang area, with a rate of 3.44 $d \cdot a^{-1}$ ($p < 0.05$), which was mainly concentrated in the central Tianshan Mountains area and the Kunlun Mountains.

Snow phenology in mountainous areas is closely related to altitude. Figure 10 shows the calculated 20-year histogram of the snow phenology trend. Areas with perennial snow cover were not included in the calculation to avoid errors in mixing seasonal and permanent snow cover. Different mountains have different snow lines, so the elevation selected for calculation was different. The Altai Mountains and Tianshan Mountains do not consider

the area above 4000 m, and the Kunlun Mountains do not consider the area above 5500 m. The trend analysis showed that the SCD trend of the Altai Mountains increased with the decrease in altitude in the altitude area less than 1500 m and increased with the increase in altitude between 1500 m and 2500 m. The trend of the SCD and SOD in the Tianshan area reached the maximum at 3000 m and was delayed by the regional SED at 3000 m. The SCD in the Kunlun Mountains area maintained a negative trend, and the SED had the largest trend between 4000 and 4500 m.

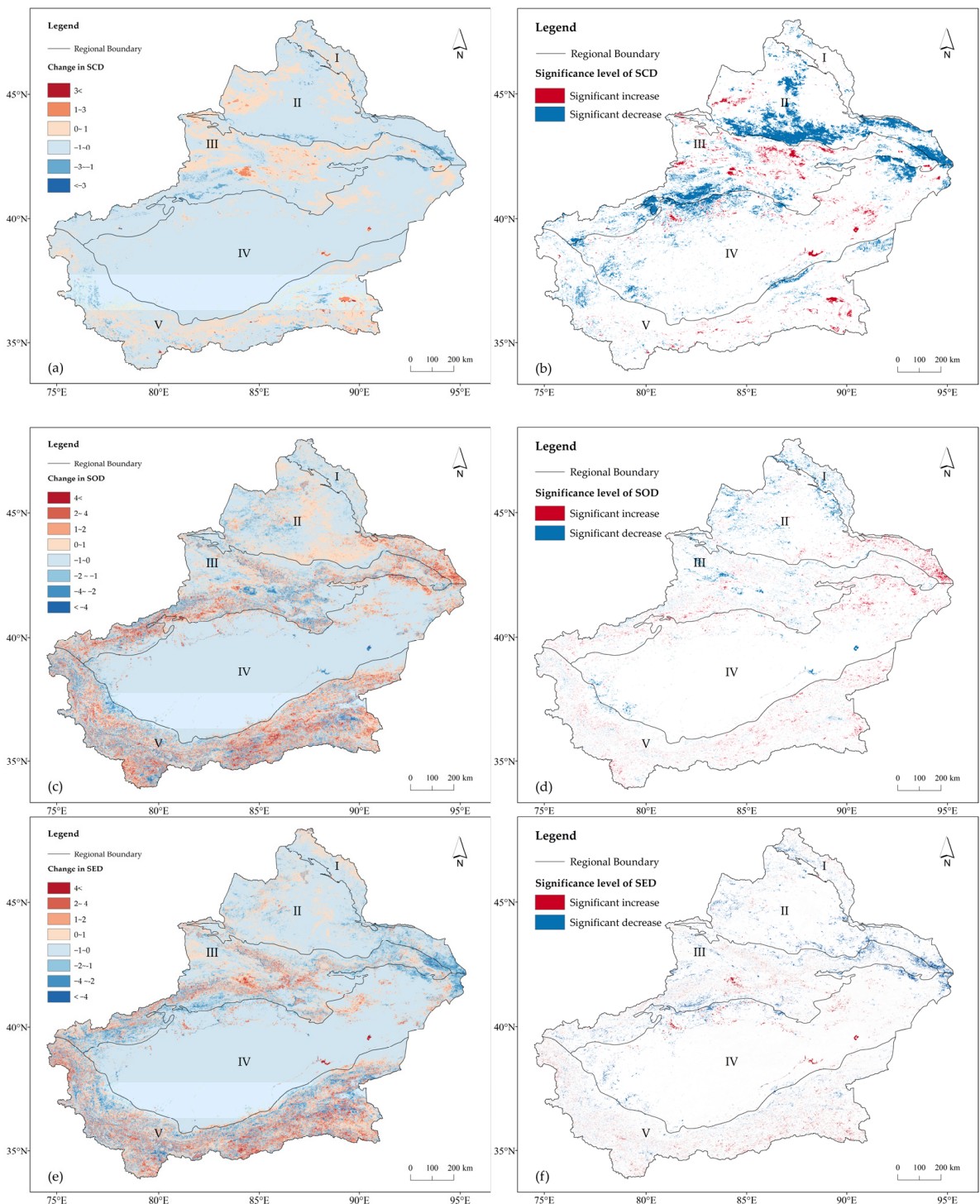

**Figure 9.** Changes and significance levels of snow cover in Xinjiang during 2001–2020: SCD (**a**,**b**); SOD (**c**,**d**); and SED (**e**,**f**).

**Table 4.** The trend changes of snow phenology and the area proportion of significant trend change.

| Regions | SCD | | | SOD | | | SED | | |
|---|---|---|---|---|---|---|---|---|---|
| | Changes Unit: d/a | Significant Decrease (Changes) | Significant Increase (Changes) | Changes Unit: d/a | Significant Decrease (Changes) | Significant Increase (Changes) | Changes Unit: d/a | Significant Decrease (Changes) | Significant Increase (Changes) |
| I | −0.21 | 3.47% (−0.88 d/a) | 0.18% (0.69 d/a) | −0.33 | 8.12% (−1.28 d/a) | 0.39% (2.42 d/a) | −0.29 | 3.87% (−4.92 d/a) | 0.83% (4.18 d/a) |
| II | −0.3 | 21.63% (−0.76 d/a) | 0.69% (0.84 d/a) | −0.07 | 3.21% (−1.44 d/a) | 1.76% (2.22 d/a) | −0.36 | 3.23% (−1.72 d/a) | 0.18% (2.42 d/a) |
| III | −0.12 | 9.67% (−0.79 d/a) | 3.03% (0.92 d/a) | −0.1 | 2.52% (−2.7 d/a) | 1.33% (3.33 d/a) | −0.22 | 3.7% (−2.46 d/a) | 1.23% (3.27 d/a) |
| IV | −0.03 | 5.74% (−0.36 d/a) | 1.07% (0.53 d/a) | 0.02 | 0.59% (−1.61 d/a) | 0.74% (1.93 d/a) | −0.03 | 1.41% (−1.64 d/a) | 0.59% (2.09 d/a) |
| V | −0.11 | 3.32% (−0.74 d/a) | 1.34% (1.7 d/a) | 0.12 | 1.2% (−4.93) | 2.1% (3.65) | −0.01 | 1.69% (−1.36 d/a) | 2.12% (1.36 d/a) |
| Southern Xinjiang | −0.08 | 5.96% (−0.8 d/a) | 1.24% (0.84 d/a) | 0.04 | 1% (−1.56 d/a) | 1.31% (2.36 d/a) | −0.05 | 1.77% (−1.79 d/a) | 1.22% (2.51 d/a) |
| Northern Xinjiang | −0.21 | 15.08% (−0.54 d/a) | 1.96% (1.05 d/a) | −0.12 | 3.7% (−3.48 d/a) | 1.36% (3.12 d/a) | −0.32 | 3.67% (−3.01 d/a) | 0.47% (3.58 d/a) |
| Whole Region | −0.12 | 8.49% (−0.67 d/a) | 1.44% (0.96 d/a) | −0.004 | 1.75% (−2.35 d/a) | 1.32% (3.0 d/a) | −0.13 | 2.3% (2.47 d/a) | 1.01% (3.44 d/a) |

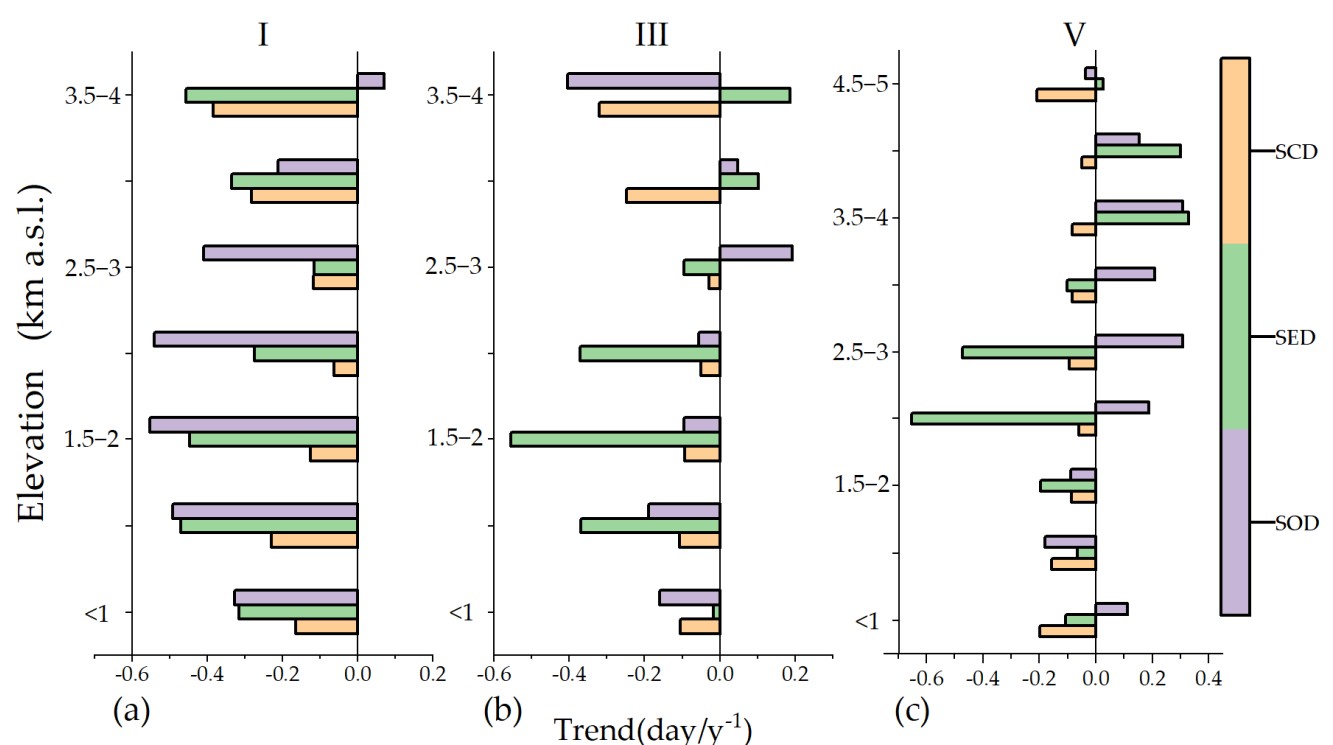

**Figure 10.** Depiction of trend of duration of snow cover phenology with elevation during 2001–2020: (**a**) Altai Mountains; (**b**) Tienshan Mountains; (**c**) Kunlun Mountains.

## 4. Discussion

In this study, a new method was proposed to extract snow phenological parameters from a MOD10A1 data set. The SCD of Wang and Xie's method was 90% consistent with the observation of the site, and SOD or SED had a good consistency, with an average of one week in advance or one week in delay, respectively. The accuracy of snow phenological parameters of our newly proposed method was compared with the real-time camera data, and it was found that SOD and SED had a difference of 0 to 3 days with the real-time

camera data. Although there was a cloud effect, the effect on SOD and SED was small. Even if some of them did have bias, the bias should be relatively constant every year. Therefore, the time series of these snow cover parameters enables us to study the trend changes of these parameters. Snow cover begins earlier in the mountains, melts later, and lasts longer, and there are fewer snow days in the basins, where snow starts late or is even delayed until the following year. Therefore, after comparison, different regions adapt different snow phenological extraction methods, which needs further evaluation and comparison. For example, one method is used to extract snow phenological parameters in mountainous areas, and another method is used in basin plain areas. It is expected that these methods of extracting snow cover parameters will enable us to study the spatiotemporal variations of snow cover in more detail.

In the context of global warming, the spatial variation in snow cover phenology in Xinjiang is consistent with the decrease in snow cover area in most parts of the Northern Hemisphere [47]. For example, Hori's study showed a weak shortening trend in the duration of snow cover in eastern Asia and western and northern North America from 1978 to 2015 [65]. Chen and his team studied the SCD in the Tianshan region of Central Asia and showed an inconspicuous upward trend during 2002–2017, but the SCD decreased, the SOD was delayed, and the SED was delayed in 38.6% of the region, mainly in the central and eastern part of the Xinjiang Tianshan Mountains [43]. This was consistent with the results obtained by Hu [66] in studying the changes of the SOD and SED in arid and semi-arid areas of Xinjiang.

Geomorphology has an important effect on the snow phenology in Xinjiang; the elevation and underlying surface type play a key role in the distribution of the snow phenology. For high altitude areas, there is a strong linear relationship between snow phenology and altitude, and every 100 m increase in the SCD requires 1.2–2.6 d (Figure 6), indicating that the change of the SCD is highly dependent on altitude. At the same time, the trend analysis showed that the response of snow phenology at high and low altitudes presented a contrast, and the mountain areas of Xinjiang maintained a negative trend in the change of altitude trend (Figure 10a). In the latitude direction (Figure 7), the SCD anomalies appeared in the Junggar Basin at 46°~48°N because the elevation at 47°N was higher than the surrounding surface. Secondly, it may be that because there are more cities between the Tianshan Mountains and the Junggar Basin, human activities are more frequent, and the SCD is less. The higher the latitude, the less human intervention, resulting in the increase of the SCD. In the longitude direction of the study area, the SCD decreased, the SOD delayed, and the SED advanced with the increase of longitude. The reason for the great change of snow phenological parameters in the longitude direction was the change of topography in the study area. The greater the longitude, the lower the average altitude was. Among them, the SCD of 86°~88°E increased, which was due to the elevation of Altai Mountain from low to high in the range of 86°~88°E.

Under the influence of different land-use types, soil surface and soil temperature, and moisture change, vegetation can reflect solar radiation. Different land-use types obtain different amounts of solar radiation, and the amount of solar radiation directly affects the temperature level, affecting snow accumulation and melting time in the region. Snow accumulation in wasteland was the latest, the SOD was later than forest land and grassland, and the SED was also much earlier than other land types, which resulted in the shortest SCD in wasteland, which was also one of the reasons for the lowest SCD in the Tarim Basin. The number of days with snow cover was the largest in forest land, and the SOD and SED were earlier and later than other land types, respectively. This may be due to the influence of factors, such as local climate convection in forest land. The forest canopy has the function of heat preservation and moisture, and the decrease of wind speed slows down the rate of snow melt, which makes the number of days with snow cover in woodland higher than other land types.

A new method for calculating snow phenology is proposed in this paper, but there is still potential for further improvement, especially in areas with perennial snow cover

and less snow cover. At the same time, the accuracy of some current cloud elimination methods is not clear, so the cloud factor is not considered in this paper. However, it is the direction of further efforts to verify the current cloud removal methods with more phenological cameras or to develop cloud removal methods with higher accuracy. Of course, before this, this method will have a certain impact on the identification of snow cover days. However, ongoing challenges, such as the impact of heavily shaded areas on snow cover judgment and scale differences between the phenological camera and MODIS products, will continue to accompany the development of this study. At the same time, the analysis of the correlation between climate and snow phenology is also a very meaningful direction of work, which needs to be explored by researchers.

## 5. Conclusions

1. We proposed a snow phenology extraction method based on MOD10A1, combined with the temperate continental climate and unique terrain conditions in Xinjiang, and proposed an algorithm suitable for snow phenology retrieval in Xinjiang. The difference between the proposed snow phenological parameters and the snow phenological parameters of the four field camera observation points was 1–3 days, and the calculated mean absolute error (MAE) and root mean square error (RMSE) values were 0.65 and 1.07, respectively.

2. According to this research, the snow cover area is mainly distributed in the Altai Mountains, the Junggar Basin, the Tianshan District, and the Kunlun Mountains, accounting for 80.72% of the total area of the study area. The area with the SCD of more than 100 days accounted for 5.15% of the total area of the study area, and the area with the SCD of less than 30 days accounted for 70.98%. The mean annual values of the SCD, SOD, and SED were 25, 342 (8 December), and 51 (8 February) in the day of year (DOY), respectively. In the mountainous areas, the SCD was in early September, the occurrence of SOD was in November and later in the Junggar Basin, the occurrence of SED was in January to June, and the occurrence of SED was in March and before in the two basins. For high-altitude mountains, it was delayed until May and beyond. Below the altitude of 2000 m, the SCD rising trend was slow with the increase in altitude, and the SCD rising trend was faster when the altitude exceeded 2000 m. The lower the elevation of the SOD, the later the SOD, and the higher the elevation, the later the SED. The snow cover phenological parameters in the mountain region showed an obvious elevation gradient. The mean gradient of the SCD in the Altai Mountains, the Tianshan Mountains, and the Kunlun Mountains is 2.6 d, 2.1 d, and 1.2 d/100 m, respectively. The snow phenological parameters fluctuate greatly with latitude and longitude. Two inflection points with large fluctuation were found in the analysis of longitude variation of phenological parameters, which may be related to the vertical zonality of the study area. The fluctuation of snow cover days of different land-use types remained stable during the study period. The snow cover days of wasteland were the least, while the snow cover days of woodland were the most; woodland was the first to enter the state of snow accumulation, and the snow cover days of wasteland were the earliest.

3. The SOD was significantly advanced in 1.75% of Xinjiang ($p < 0.05$), and the advance rate was about 2.35 $d{\cdot}a^{-1}$. The SOD was significantly delayed in 1.32% of the region ($p < 0.05$) at a rate of about 3 $d{\cdot}a^{-1}$. The SED was significantly advanced in 2.38% of Xinjiang ($p < 0.05$), and the advance rate was 2.47 $d{\cdot}a^{-1}$, while the SED was significantly delayed in 1.01% of the Xinjiang region, and the advance rate was about 3.44 d $d{\cdot}a^{-1}$. The SCD decreased significantly in 8.49% of Xinjiang, in the Junggar Basin, and in southern Xinjiang, and the shortening rate was about 0.67 $d{\cdot}a^{-1}$. The SCD in 1.44% of Xinjiang showed a significant lengthening trend, and the lengthening rate was about 0.97 $d{\cdot}a^{-1}$. For the whole of Xinjiang, the SCD showed an insignificant decreasing trend at 0.12 $d{\cdot}a^{-1}$, which was mainly related to the advance of the SED.

**Author Contributions:** Conceptualization, Y.M. and J.L.; methodology, Q.W. and Y.M.; software, Q.W. and Y.M.; validation, Q.W. and Y.M.; formal analysis, Q.W.; investigation, Y.M.; resources, Q.W., Y.M. and J.L.; data curation, Q.W.; writing—original draft preparation, Q.W.; writing—review and editing, Y.M. and J.L.; visualization, Q.W.; supervision, Y.M.; project administration, Y.M. and J.L.; funding acquisition, Y.M. and J.L. All authors have read and agreed to the published version of the manuscript.

**Funding:** This research was funded by the Third Xinjiang Scientific Expedition Program (Grant No. 2021xjkk1400), the National Natural Science Foundation of China (Grant No. 42071049), the Natural Science Foundation of Xinjiang Uygur Autonomous Region (no. 2019D01C022), Xinjiang Uygur Autonomous Region innovation environment Construction special project & Science and technology innovation base construction project (PT2107). Tianshan Talent-Science and Technology Innovation Team (2022TSYCTD0006).

**Data Availability Statement:** Not applicable.

**Acknowledgments:** The authors would like to thank the three anonymous reviewers for their valuable comments and the editor for her help with this article.

**Conflicts of Interest:** The authors declare no conflict of interest.

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
