# Peer review of "Snow Cover Phenology in Xinjiang Based on a Novel Method and MOD10A1 Data"

_remotesensing, doi:10.3390/rs15061474_

Round 1

Reviewer 1 Report

1.    The introduction needs to be expanded.

2.    Please provide a bit more big-picture motivation of how your analyses benefit society and how they have evolved over the past decade. However, from my point of view, the article does not provide a sufficiently thorough review of the issue under study. There are good references for the study techniques, but the paper is missing a "big-picture" introduction with some references in my opinion. I suggest that the authors should do a better analysis of the literature. It seems that the bulk of the text is a sort of compilation of statements in the individual articles cited. It would be better, I think, to extract ideas from individual articles and tie them together into a more fluid and conceptually homogeneous text. As it is, the text looks rather clumsy.

3.    Research gaps, objectives of the proposed work should be clearly justified before the problem formulation section. This paper includes some little useful information and the main objectives of the study is not well defined. Problem statement is not clear and the objectives are obscure. Furthermore, the paper lacks a very clear and good justification for what is new and innovative about this case or this approach.

4.    WHY (I). Altai Mountains (II). Junggar Basin (III). Tienshan Mountains (IV). Tarim Basin (V). Kunlun Mountains are specially listed in your study?

5.    In the discussion of the relationship between snow cover phenology and land use, the time and processing method of land use data need to be clarified.

6.    In section 3.2.2, the change rate of the regions that have changed significantly in snow cover phenological parameters needs to be added in table 4.

7.    More detail should be supplied in the comparison of the three snow phenology extraction methods shown in Figure 3.

8.    Enlarge the legend size in Figures 5 and Figures 9.

9.    In the discussion section, it should be explained the applicability of the extraction method of snow phenology.

Reviewer 2 Report

Journal: Remote Sensing

Manuscript title: Snow Cover Phenology in Xinjiang Based on a Novel Method and MOD10A1 Data

Authors: Qingxue Wang, Yonggang Ma, Junli Li

Review:

The manuscript is a very interesting description of snow cover duration, snow onset date and snow end date for 17 land cover types depending on elevation and geographical latitude and longitude in 5 regions of Xinjiang (China). The working method is very well explained. The advantage of the work is checking the accuracy of MOD10A1 method by using four field camera observation sites. The accuracy turned out to be very good.

I recommend the manuscript to be published in the journal Remote Sensing. However, some clarifications and additions are needed:

·       the title: in my opinion the expression “snow cover phenology” jest debatable. Phenology is the study of natural phenomena that recur periodically in plants and animals and of the relationship of these phenomena to seasonal changes and climate. I suggest to change the title to “Snow cover in Xinjiang…”

·       line 25-26: “The spatial–temporal variation in snow phenology was highly altitude dependent.” The conclusion is rather obvious. Supplement it, please, with specific values describing the examined dependencies.

·       line 34: “…were non-significant”. What was non-significant: “snow cover parameters” or their trends?

·       line 128: it is necessary to add in the text what is the height above sea level of the studied areas; emphasize the large difference in altitude (-192, 7538 m a.s.l.).

·       lines 265-277: The conclusion on changes in snow cover with altitude is rather obvious. I would suggest to add more numerical information in the text describing the changes of SCD, SOD and SED per 100 m of elevation.

·       Fig. 7a: what is the cause of unusual changes of SCD in region II with latitude of 46-48°?

·       Fig. 7c: what is the cause of unusual changes of SCD in region I with longitude of 86-88°?

·       lines 293-294: “Figure 7(c,d) show that the SCD in the study area decreased with the increase in longitude, while the SOD was delayed, and the SED advanced.” What is the reason for such changes of SCD, SOD and SED with longitude? Explain, please.

·       line 300: “… of snow ablation in this region.” I suggest to change to: “… of snow accumulation and ablation in this region”;

·       lines 299-304: please, add to the text the information about the average differences in the number of days of SCD, SOD and SED between forest, grasslands and barren areas.

·       line 335 and bellow: in my opinion the 20-year period is far too short to analyse trends and draw conclusions on them;

·       line 375: “… the SCD of the Altai Mountains…” change to “… the SCD trend of the Altai Mountains…”

·       lines 403-406: detailed description; move, please, the text to the subsection 3.2.2.

·       lines 416-418: why SOD is earlier in forest land? SOD does not depend on the land use, but on the direction of air advection and the first snowfall, which may occur on the same day over the forest and forestless area.

·       lines 441-450: add, please, selected numerical data describing changes in SCD with elevation, latitude and longitude.

Minor remarks:

·       line 25 and line 223: “…0.65 and 1.07” respectively (?)

·       line 30: “… 341.58 and 50.84 in day of year…”  Replace, please, the values in the text with a specific date.

·       lines 116-117 and 389-390: “from the State Key Laboratory of Desert and Oasis Ecology, Xinjiang Institute of Ecology and Geography”; the name of the institution is not needed; I suggest you to delete it.

·       lines 145-146; line 213: what was the period of camera observations: “2017 to 2019” (as in lines 145-146) or “2017 to 2020” (as in line 213)?

·       Table 4: which regions are included in Southern Xinjiang and which in Northern one?

Reviewer 3 Report

The article "Snow Cover Phenology in Xinjiang Based on a Novel Method and MOD10A1 Data" describes a novel derivation of phenological parameters to detect changes in the snow cover season (duration, onset and melt). In a pixel-based approach, the MODIS snow product MOD10A1 is examined for the Xinjiang region and validated with in situ camera recordings. The investigation took place separately for 5 regions and for the entire area. There is also an analysis of snow cover variability for different land cover classes and a trend analysis based on MK-Test and Sens's slope.

The introduction leads well to the topic and the (very detailed) literature study also offers a very good introduction to the topic. You should only pay attention to a uniform citation method (see below). However, a paragraph on the objective of this paper and the formulation of research questions is missing and should be added.

The chapter on material and methods is clear and almost complete, minor suggestions for improvement are attached below. However, what is missing is the description of how to get from the scaled NDSI values in the MOD10A1 layer "NDSI_snow_cover" to the binary classification into "snow" and "snow-free" (as shown in Figure 3). The same applies to the results section, although the section on the Mann-Kendall test contained there should be incorporated into the "Material and Methods" section.

In my opinion, some points are still missing in the discussion chapter. The variability is mainly attributed to geographic features, but a critical examination of the data used is missing here. How, for example, are the data gaps caused by clouds etc. treated and what falsifications does this lead to? In addition, the spatial resolution of MODIS in heavily reliefed areas should be discussed. I also think that the new method leads to a general overestimation of the SCD, which should at least be addressed.

I have one general comment about the illustrations: the font size is often very small and illegible, so the entire manuscript should be revised accordingly.

Here are more detailed suggestions for improvement:

Line 16: I would recommend using the term "earth observation" instead of "space technology".

Line 22: It is a bit confusing when you speak of daily data, but they only record from 2017-2020. Maybe you should speak of "daily data during an overlapping period from 2017-2020".

Line 30: Remove the "d" from "24.81d".

Line 35: You should also iclude a keyword for the trend analysis you performed (either "MK-Test" or "trend analysis").

Line 75 ff.: Please keep the MDPI numeric referencing style, you should avoid naming the authors (also sometimes they are in capital letters, sometimes not). The numeric citation is sometimes placed at the end of the sentence, sometimes right after the author's name. This should at least be consistent. If you choose to leave it as is, I would recommend using "author [number reference]".

Line 115: It is not clear what you mean with "... of the hydrological year (DOY)" since the abbreviation DOY is for "Day of Year". You should also define the hydrological year (start/end). If the day reference refers to the hydrological year, should that perhaps be labeled "day of hydrological year (DOHY)"?

Line 135: Very nice overview map! But could you show the north-south border in a more visible color. And I would personally prefer a color scaling of the DEM.

Line 139: Please state which tiles you used. A citation from Dorothea Hall (https://doi.org/10.1016/S0034-4257(02)00095-0) how created  the MOD10A1 product would also be important.

Line 166 (Table 2): In the Description "n" is described, in the SCD Formula this is written in capitals. In the description it is also missing what the subscripts 1 and 2 stand for? Also the SCD methods seem to be the same for Wange and Xie and Gao (in the latter, the formula is described only in words). And is this information missing in the Chinese method? In the first method by Wang and Xie, it is also not clear how the binary decision between snow and snow-free comes about. If, for example, an NDSI threshold value of > 0.4 is taken, this corresponds to a snow-covered area of > 50% (i.e. the same as the definition of the Chinese method). I think the table as a whole should be revised.

Line 186 f.: I don't understand this sentence. "NDSI_snow_cover" is just a layer of the daily MODIS data set. The positive NDSI values are scaled between 0 and 100 for the original values from 0 to 1 (since the NDSI can have a value between -1 and +1, but only positive values can indicate snow). It is also not mentioned how the binary classification von "snow" and "snow-free" was done.

Line 213: Please explain in more detail how the very small-scale camera recordings were scaled to the 500 m MODIS pixels.

Line 243 (Figure 4): Please change the X-axis label to "Snow cover [%]".

Line 262 (Figure 5): Please increase the font size of the legends.

Line 310: The entire section 3.2.1 belongs in the materials and methods section, not in the results section.

Line 311 ff.: I am missing the citations of the Mann-Kendall Test and the Sen's slope (https://doi.org/10.2307/1907187, https://doi.org/10.1080/01621459.1968.10480934), please insert.

Line 321 ff.: Please cite at least one study.

Line 363: The inscriptions in the subplots of Figure 9 are far too small and illegible. Please improve!

All the best!

Reviewer 4 Report

In this paper, the authors used a new method to calculate SCD, SOD and SED in Xinjiang and analyzed the climatology and variations in snow cover phenology.  However, several issues need to be corrected before publication

Specific comments:

1.     Line 30: the values of mean annual SCD, SOD and SED should keep an integer. And I suggest the average SOD and SED should be shown as the months to understand clearly.

2.     Line 52: delete the subtitle

3.     Line 62: when SCD, SOD, and SED appear for the first time in the text, use their full names instead of abbreviations.

4.     Line 71: disappearàappear?

5.     The authors cite references in the wrong way. Please refer to the journal format requirements. A paper written by three or more than three authors should cite “XX et al.” please check and revise the whole manuscript.

6.     Line 84: delete “the duration of”

7.     In the introduction section: the authors should explain why do this research. They just show the previous research. What are the shortcomings of these studies, and what new findings in this new study will fill the gaps? These need to be stated in this section.

8.     Line 125: 73°40'~96°18' east longitudeà73°40'~96°18' E

9.     Line 126: 34°25'~48°10' north latitudeà34°25'~48°10' N

10.   Line 136: delete “Sketch of”

11.   Line 151: why not use 500 m DEM data to match?

12.   In the 2.3.4 section: I suggest the authors add the definition of snow cover year. Snow cover year is a complete period with snow accumulation, stable and melt period. In the current manuscript, we do not know when snowpack starts, in the last year or the current year. The same as snowpack ends.

13.   Line 270: the inflexion point in the Tienshan Mt. appears at 3500 m, not 4500 m. Please check.

14.   Line 272: what does perennial mean? whether include glaciers? why SOD (SED) of perennial snow cover showed the later (earlier) trend? SCD increased with an elevation not decreased, please check.

15.   Line 287: add N after 40°

16.   3.2.1 section: move this part to the method section

17.   Lines 389-390: delete the affiliation

Reviewer 5 Report

The study analyses snow cover duration and snow cover onset and end date and their changes in the Xinjiang region. The analysis of snow phenology is based on daily MODIS snow data. The results indicate that the snow phenology is altitude dependent. In most areas, the temporal changes in snow phenology are statistically non-significant.

The topic of assessment of snow phenology parameters from satellite observations is within the scope of the journal. However, the presentation and demonstration of the novel scientific contribution of the manuscript need significant improvement before I can suggest it for publication. My main critical comments are related to the following:

1) The clarity of the presentation needs to be substantially improved. It includes both the general structure of the manuscript and the formulation of the novel contribution. The manuscript will benefit from a clear formulation of the objectives and scientific novelty in the Introduction part. It is important to present clearly what the current status of the research is and what the research gaps are. In its current form, the manuscript reviews numerous studies investigating changes in snow cover, without a clear formulation of the research objectives and which aspects are new and why. From reading the methods, it seems that the main focus is on proposing a new method for estimating snow phenology parameters. Still, the Introduction does not review the existing approaches nor formulates why new approaches are needed and which aspects of existing approaches will be improved. The methodological section presents and compares two existing approaches, but there is much more literature (and methods) used in the past.

The clarity of the presentation can also be improved by revising the structure. I suggest presenting all aspects of the applied methods in the Methodology section. In its current form, some parts of the methods are presented in the Results (e.g. the trend analysis). Also the clarity of Table and Figure captions needs to be improved. The captions do not clearly explain what exactly is presented, the abbreviations used are not explained, 

2) The assumptions of the proposed method are not clearly communicated and justified. My main concern is the impact of cloud coverage on the results. How are cloud pixels handled in the evaluation? Are the results sensitive to the occurrence of cloud pixels? Another question is the snow cover mapping based on NDSI values from MODIS. Are all pixels with an NDSI value larger than 0 considered as snow-covered? The previous versions of MODIS snow cover datasets used some NDSI threshold for the snow cover mapping. Can this be justified/validated by using camera observations?

3) If the main aim is to propose a new method, it will be interesting to see how accurate the new method is compared to other existing approaches. It will be very interesting to use the camera observations for such verification.

4) Most of the results reads more as the evaluation of the snow phenology in the study region. Here I'm confused. The results need to be closely linked with the main research aims. The manuscript in its current form reads more as a case study that evaluated the snow phenology and its changes in the study region, not as documentation that the newly proposed method is more accurate and informative than existing approaches.

Addressing of these comments can improve the message and presentation of the novel scientific contribution of the paper.

Round 2

Reviewer 1 Report

Authors have responded clearly to all my comments and updated the manuscript accordingly. Thus, I find it suitable for publication.

Author Response

 We much appreciate the reviewer’s suggestion. 

Reviewer 4 Report

I appreciate the authors' time and effort to improve manuscripts and for providing detailed responses to the comments. I am happy with the author's reply to my previous comments. I would like to suggest this manuscript for publication before minor revision.

1.      Lines 60-61: I suggest that this sentence be changed to “The result shows that SCD has decreased at the global mountains [ref.].

2.      Line 90: snow cover extent appears twice

3.      Lines 99-110: Two objectives. There is a repeat statement of the objective. Simplify, please.

4.      Line 114: delete “east longitude” and “north latitude”

5.      Line 116: the unit of altitude should be meter, not km

6.      Lines 160-185: I suggest moving the trend analysis method section after the 2.4 Accuracy section according to the order of the Results section.

7.      Lines 241-243: I asked the authors how to define the snow cover year in the last manuscript. While in this new text, I think the authors misunderstand my meaning. Snow cover year concludes snow accumulation, stable period and melt period, SOD comes first and is followed by SED. They should appear in different calendar years. However, as the authors defined from 1st Jan. to 31st Dec., the 98th date is the SED and the 308th date is the SOD, SED comes first. This is clearly unreasonable. 98th should be the SED of the last snow cover year and 308th be the SOD of the current year. They belong to the two different snow cover years.

8.      Table 3: What does the number mean below every station name? altitude? If yes, the unit should be the meter.

9.      Lines 320-322: This sentence is to explain the reason for what? 1) for the elevation gradient? If yes, it should be moved after “(Figure 6c)”. 2) or explain the inflection point? Does it mean there is perennial snow or glacier causing an elevation inflection point? It is confusing.

10.   Lines 450, 458 and 459: insert N or E after the number.

11.   Line 454: What does SOD decreased mean? Advanced?

Reviewer 5 Report

dear authors,

i really appreciate your willingness to address the comments raised in the reviews and to improve the manuscript. however, i have still two suggestions (one general and one specific which (i think) are worth considering before the publication:

1) I believe the formulation and presentation of the scientific novelty can still be improved. i want to highlight that the main scope and aim of the journal is to "publish novel / improved methods / approaches and / or algorithms of remote sensing to benefit the community, open to everyone in need of them". The formulation of the objectives needs to be better balanced with these aims and precisely formulate the novel scientific contribution, i.e. some new understanding/knowledge which goes beyond a case study analysis of snow cover variability in a selected region. Suppose the main research objective is to develop a new method for extraction of the snow cover characteristics. In that case, it is important to clearly indicate why a new approach is needed and how and why it outperforms the existing methods. This is partly presented in the manuscript, but in its current form, it is mixed with an extended evaluation of how the snow cover characteristics vary and change in the study region. Suppose the objective is more to understand the changes in the variability of snow cover characteristics (by using remote sensing products). In that case, the analysis needs to go beyond presenting only the variability and to attribute and understand the factors which are causing these changes. Both objectives are interesting and relevant, but in its current form (of the manuscript), they are mixed and need to provide a coherent story. Moreover, all these potentially novel aspects are hidden by presenting the extended case study evaluation of the variability in the study region.

2) Please carefully check the Table and Figure captions. It is an advantage for the reader if the captions are self-explanatory (using the abbreviations here makes understanding difficult).
